# Depression Associated with Reduced Heart Rate Variability Predicts Outcome in Adult Congenital Heart Disease

**DOI:** 10.3390/jcm10081554

**Published:** 2021-04-07

**Authors:** Mechthild Westhoff-Bleck, Lars H. Lemke, Jan-Marc S. Bleck, Anja C. Bleck, Johann Bauersachs, Kai G. Kahl

**Affiliations:** 1Adult Congenital Heart Centre, Department of Cardiology and Angiology, Hannover Medical School, D-30625 Hannover, Germany; Lars.H.Lemke@gmx.de (L.H.L.); Anja.Bleck@stud.mh-hannover.de (A.C.B.); Bauersachs.Johann@MH-Hannover.de (J.B.); 2Department of Psychiatry, Social Psychiatry and Psychotherapy, Hannover Medical School, D-30625 Hannover, Germany; Jan-Marc.Bleck@stud.mh-hannover.de (J.-M.S.B.); Kahl.Kai@mh-hannover.de (K.G.K.)

**Keywords:** depression, heart rate variability, prognosis, heart failure, arrhythmias, mortality, hospitalization, adult congenital heart disease

## Abstract

In adult congenital heart disease (ACHD), major depressive disorder (MDD) represents a frequent comorbidity. In non-CHD, adverse outcome is predicted by MDD and heart rate variability (HRV), whereas in ACHD their prognostic relevance is unknown. We prospectively evaluated 171 patients (age 35.6 ± 11.4 years; male 42.7%, mean observation time 54.7 ± 14.9 months). Binary regression analysis calculated the association between MDD and HRV. Cox proportional survival analysis estimated their impact on decompensated heart failure and all-cause mortality (HF/death), supraventricular and ventricular tachycardia (SVT/VT), and hospitalization due to unexpected cardiac causes. Exclusively MDD with moderate/severe symptoms showed significantly lower HRV as derived from frequency-domain analysis (Symindex) (*p* = 0.013). In multivariate Cox regression analysis, patients stratified according to the lower quartile of the Symindex comorbid with MDD (n = 16) exhibited poorer prognosis regarding HF/death (Hazard Ratio (HR): 7.04 (95%CI:(1.87–26.5)), SVT/VT (HR: 4.90 (95%CI:1.74–9.25)) and hospitalization (HR: 3.80 (95%CI:1.36–10.6)). An additional independent predictor was N-terminal pro-B-type natriuretic peptide elevation (*p* < 0.001), indicating advanced HF and heart disease complexity (*p* < 0.001). Autonomic nervous system dysfunction measured by altered HRV is considered to be one of the pathways linking MDD and adverse outcomes in cardiac diseases. Our results exceed the existing literature by demonstrating that MDD with decreased HRV is associated with poorer prognosis in ACHD.

## 1. Introduction

In adult congenital heart disease (ACHD), major depressive disorder (MDD) represents a frequent comorbidity, being present in up to 33% of patients [1,2,3]. In numerous non-congenital heart disease conditions, depression predicts poorer outcome in terms of an increased risk of hospitalization due to any cardiac event, including the worsening of heart failure (HF) and all-cause mortality [4,5,6,7,8]. Depression also promotes pro-arrhythmic effects, contributing to an increased risk of incident supraventricular tachycardia, the worsening of the clinical course in chronic atrial fibrillation and incident ventricular tachycardia [9,10,11].

The underlying pathophysiology points to shared common pathways observed in MDD and cardiovascular diseases. In particular, in HF, bidirectional heart brain interactions involving higher cortical and brain stem function contribute to autonomic dysfunction, inflammation and an increased activity of the hypothalamic–pituitary–adrenal axis. These systems modulate multiple cardiopulmonary reflexes and neuroendocrine signalling, responsible for the regulation of organ perfusion, blood pressure, heart rate and fluid control, and the modulation of cardiac output and ventilation [11,12,13].

In particular, the alteration of the autonomic system with subsequent reduced heart rate variability (HRV) plays a central role in both HF and MDD, representing an independent predictor of adverse outcome in a variety of underlying cardiac conditions [4,5,6,7,8,9,10,14,15,16].

HRV is regarded as a biomarker of the autonomic nervous system. Physiologically, the heart rate is not a rigid variable. It rather reflects the functional adoption to the demands of environmental and psychological challenges embedded in a variety of regulating mechanisms, including bidirectional modulation of the higher cortical and brain stem function, the autonomous nervous system with its sympathetic and parasympathetic properties, baroreceptors and chemoreceptors’ activity and respiratory regulation [17,18]. Loss in variation reflects a pathological state associated with adverse outcome [4,5,6,7,8,9,10,14,15,16,17,18,19].

In ACHD, reduced HRV has been reported in diverging congenital cardiac defects such as Fallot-Tetralogy, systemic right ventricles, Fontan circulation, after operative ventricular septal defect closure associated with bundle branch block and cyanotic heart disease; however, data with respect to outcome are lacking [20,21,22,23,24,25].

On the other hand, there is some evidence that depression might influence outcome in ACHD. However, applied inconsistent assessment methods in diagnosing MDD and variable endpoints limit the information regarding prognosis [26,27,28].

To date, it is completely unknown whether MDD associated with reduced HRV adversely predicts prognosis in ACHD. This prospective study aimed to analyse the impact of reduced HRV in depressed ACHD on future decompensated HF, all-cause mortality (HF/death), supraventricular and ventricular arrhythmias (SVT/VT) and unexpected cardiac causes of hospitalization.

## 2. Methods

The PsyConHeart study is an ongoing project evaluating the importance of mental illness in ACHD [1,29,30]. The present cohort represents a subgroup of the initial cross-sectional study, which investigated the prevalence of psychiatric comorbidities in ACHD [1]. The diagnosis of depression was based on a psychiatric clinical interview according to the Diagnostic and Statistical Manual of Mental Disorders-IV (DSM-IV) -criteria. The Montgomery–Åsberg Depression Rating Scale assessed symptom severity. A detailed description of the patients’ inclusion has been reported previously [30].

The subgroup included 171 patients, who participated in HRV measurements. None of the patients presented with pacemaker rhythm or supraventricular or ventricular tachycardia. Only one patient was lost during follow-up.

Cardiological and psychological assessment was performed on the same day. Heart rate variability, blood tests and electrocardiograms (ECG) were taken prior to psychological assessment.

Disease severity was assessed according the Warnes classification [31]. A detailed description of the underlying cardiovascular anomalies is presented in Table 1.

Baseline evaluation included HRV analysis and psychological assessment. Epicardial fat was measured echocardiographically, as described previously [29]. The majority of patients presented with complex congenital lesions. Most of the patients had undergone prior cardiac surgery. At inclusion, 79.9% of the patients had a minimum of one cardiac operation.

Medical treatment included antidepressant agents (5.8%), β-Blocker therapy (32.2%), mineralocorticoid receptor antagonists (7%), angiotensin-converting-enzyme inhibitors or angiotensin II receptor blockers (40.9%), diuretics (10.5%) and anticoagulation (21.6%). Details of the clinical characteristics are presented in Table 2.

The endpoints were HF, all-cause mortality (HF/death), sustained supraventricular and ventricular tachycardia (SVT/VT) and unexpected hospitalization due to cardiac causes. HF was defined as symptoms at rest and cardiac decompensation with peripheral and/or pulmonary oedema, pleural effusion and/or ascites.

The study was approved by the ethics committee of Hannover Medical School (No. 6455/13). All participants gave written informed consent. The study follows the ethical standards of the Declaration of Helsinki.

### 2.1. Measurement of Heart Rate Variability

The evaluation of HRV was derived from short-term analysis in an upright position. Artefacts were removed prior to automatic analysis.

HRV analysis is based on different aspects of heart rate variation. Parameters of time-domain analysis calculate parameters with respect to beat-to-beat interval distribution. In contrast, frequency domain analysis operates within different frequency bands calculated by Fast Fourier Transformation or autoregressive modelling, which estimate continuous changes in the spectral power distribution.

Included parameters were the standard deviation of the interbeat intervals (SDNN), the root mean square of difference of successive interbeat intervals(RMSSD) and the calculated geometric measure of the integral of the density of the interbeat interval histogram divided by its height (Triangular Index). As the frequency-domain measurement, the Symindex calculated the logarithmically transformed HRV ratio derived from the normalized power spectral density of the low frequency band and the normalized power spectral density of the high frequency band [18].

### 2.2. Statistics

Included baseline characteristics were age and sex. Variables providing prognostic relevant information in ACHD were New York Heart Classification (NYHA-class), disease complexity, N-terminal pro-brain natriuretic peptide (NT-proBNP), inflammatory markers, renal and liver function, oxygen saturation, and the ECG parameters QRS duration and the heart rate corrected QT-duration (QTc) calculated with the Bazett formula. Additionally, MDD associated factors such as Body Mass Index (BMI), glycolized haemoglobin levels (HBA1c), low density lipoproteins (LDL) and epicardial fat were included in the analysis. Epicardial fat thickness represents a particular finding in depressed patients with ACHD [29].

ANOVA with Bonferroni correction analysed differences in continuous variables between the groups without depression and depression with either mild or moderate/severe symptoms; the Χ-square test calculated differences in categorical variables. Continuous variables are presented as mean and standard deviation, categorical variables as absolute numbers or percentage distribution.

Binary regression analysis estimated the relationship between depressive state and baseline clinical characteristics. Univariate analysis was performed in all data. Multivariate analysis with stepwise backward Wald statistic calculated independent predictors from significant univariate variables (*p* < 0.08). To exclude the potential confounding impact on clinical characteristics and biomarkers, we also adjusted for age and sex. NYHA-class and NT-proBNP measurements were closely interrelated. Therefore, multivariate analysis only included the most significant parameter.

The impact of HRV and depression on HF/death, unexpected hospitalization due to cardiac causes and SVT/VT was estimated in different models, including either the lower quartile of the Symindex (<−0.1375) or moderate/severe MDD alone or a combination of both parameters, including all patients diagnosed with MDD and HRV within the lower quartile of the Symindex. These groups were compared to all other patients independent from depressive state because patients without depression and depression with mild symptoms did not differ regarding adverse events. Cox proportional survival analysis calculated the risk on outcome. Multivariate analysis included the raw data significant in univariate analysis and adjustment for age, sex, β-Blocker therapy and a history of SVT/VT.

Receiver operating characteristics (ROC) analysis calculated sensitivity and specificity. Positive and negative predictive values are provided.

The statistical analysis was performed with IBM^®^ SPSS Statistics Version 26.

## 3. Results

### 3.1. Predictors of Heart Rate Variability

From the 48 (28.1%) patients with depression, 28 presented with moderate/severe symptoms (58.3%). The comparison between patients without MDD versus depressed patients with mild symptoms only revealed higher epicardial adipose tissue and higher values of applied self-rating scales in mildly depressed patients. In moderate/severe MDD, values of epicardial adipose tissue and depression scores were even increased compared to both patients without MDD and those with mild symptoms (Table 2).

Patients with MDD with moderate/severe symptoms had more complex congenital heart defects and a higher degree of heart failure symptoms. The number of operations was similar. Epicardial fat was elevated, but BMI and the proatherosclerotic factors HBA1c, lipid levels and high-sensitivity C-reactive protein (HsCRP) were comparable. Hypertension was equally distributed in all groups.

From all evaluated HRV parameters, exclusively the frequency domain measurement, the Symindex, gained significant results. In univariate binary regression analysis, ACHD comorbid with MDD showed a significantly lower Symindex (*p* = 0.048) (Table 1). The analysis with respect to the symptom severity of MDD calculated a particularly reduced HRV in MDD with moderate/severe symptoms (Figure 1).

In multivariate analysis, the Symindex remained as an independent predictor (*p* = 0.013) of moderate/severe MDD, even after adjustment for sex, age and disease complexity (Table 3).

### 3.2. Adverse Events

Follow-up was completed in 170 patients. The mean observation time was 54.7 ± 14.9 months. In the total cohort, eight patients developed heart failure, and eight patients died. From the eight deceased patients, four presented with a prior episode of decompensated heart failure/NYHA IV. SVT/VT occurred in 31 patients—only three of them experienced ventricular arrhythmias. Forty-six patients required hospitalization due to unexpected cardiac causes.

### 3.3. Adverse Events Related to Reduced HRV and to Reduced HRV Comorbid with MDD

Cox proportional survival analysis calculated the impact of HRV and MDD on survival. In univariate analysis, patients presenting within the lower quartile of the Symindex (<−0.1375) had a significantly higher incidence of HF/death (*p* = 0.017) and arrhythmias (*p* = 0.012), whereas hospitalization was not significantly different. MDD presented with a significantly higher incidence of HF/death (*p* = 0.019). In multivariate analysis, neither the single parameter MDD nor the Symindex reached significance.

The group of patients with reduced HRV comorbid with MDD (HRV/MDD) included a total of 16 patients: 13 of them had moderate/severe symptoms and 3 had mild symptoms. In univariate Cox proportional survival analysis, HRV/MDD presented with a higher incidence of hospitalization (*p* = 0.024), arrhythmias (*p* = 0.002) and the combined endpoint heart failure and death (*p* = 0.006). Other predictors in univariate analysis were associated with underlying congenital heart disease severity, such as NYHA-class, NT-proBNP elevation and disease complexity. Their relation to adverse outcome showed higher significance levels. In contrast, prior cardiac surgery predicted hospitalization and arrhythmias, but not HF/death.

In multivariate analysis, HRV/MDD remained as a single independent predictor in all evaluated risk groups. However, NT-proBNP elevation and disease complexity persisted as the most significant single predictors (Table 4). Even after adjustment for age, sex, β-Blocker therapy and a history of arrhythmias, reduced HRV/MDD independently predicted poorer prognosis.

In our cohort, HRV/MDD represented a particular high-risk group regarding outcome. Calculated specificity was high, whereas sensitivity was poor. In HF/death, sensitivity was 0.25, corresponding to a specificity of 94.8. In arrhythmias (0.438/0.844), values were similar. Cardiac causes of hospitalization exhibited low sensitivity (0.563) and moderate specificity (0.756). All groups presented with high negative and low positive predictive values, being 0.92/0.25 in heart failure/death, 0.935/0.225 in arrhythmias and 0.935/0.225 in cardiac causes of hospitalization, respectively, indicating that the absence of reduced HRV even in the presence of depression represents a low risk population regarding adverse outcome (Figure 2).

## 4. Discussion

This is the first prospective study evaluating the impact of reduced HRV in the context of MDD (HRV/MDD) on outcome in ACHD. Our data indicate that MDD with reduced HRV is prognostically relevant with respect to HF/death, hospitalization and SVT/VT. MDD was diagnosed by mental health professionals. Patients characterized by reduced HRV/MDD represent a particular high-risk group regarding adverse outcome.

In contrast to the combined parameter, in univariate analysis HRV predicted all three endpoints exclusively, whereas MDD only gained significance in HF/death. In addition to the combined parameter HRV/MDD, adverse events were associated with higher pre-existing cardiac damage reflected by NT-proBNP elevation and congenital lesions of great complexity.

### 4.1. Heart Rate Variability in Adult Congenital Heart Disease

In ACHD, prior studies evaluated HRV in a variety of homogenous disease entities using various time domain and/or frequency domain parameters derived from short ECG analysis or 24 h Holter monitoring [20,21,22,23,24,25]. These studies reported a relation between reduced HRV and a history of arrhythmias, atrial dilatation, ventricular septal defect closure in association with right bundle branch block, reduced ventricular function and impaired exercise capacity, indicating that HRV relates to the underlying congenital heart defect and its postoperative and long-time sequela. Thus far, our data from univariate analysis match these observations. None of the above-mentioned studies addressed the prognostic role of HRV in ACHD. As described, we could only prove the prognostic relevance of HRV in univariate, but not in multivariate analysis.

HF and MDD share common pathways involving the autonomic nervous system [11,12,13,17]. In the present cohort, MDD with moderate/severe symptoms independently predicted lower HRV. This raises the question whether MDD together with HRV might impact outcome.

### 4.2. Prognostic Value of Reduced Heart Rate Variability and MDD

At present, almost nothing is known regarding the relation between HRV and depression in ACHD. Only one study evaluated the impact of depression on HRV and baseline clinical characteristics in a cohort of cyanotic congenital heart disease [25]. Moon et al. reported that in univariate analysis, depression was associated with the logarithmically transformed high-frequency and low-frequency band and the standard deviation interbeat intervals, respectively. In contrast, at baseline we could only find a relation between depression and the logarithmically transformed HRV ratio. Time domain parameters did not retrieve consistencies. Different patient selection including an inhomogeneous group of patients with a variety of congenital lesions in our cohort might contribute to these diverging results. Moreover, we performed a short time analysis in an upright position, which excludes the influence of the circadian rhythm on HRV. In particular, the disturbances of the normal nocturnal heart rate behaviour are regarded as valuable prognostic information [18,19]. Despite an expected greater predictive power in 24-h measurements, our results and data derived from non-congenital heart disease ascertain the prognostic value of short time frequency-domain parameters regarding HF/death and SVT/VT [18,32,33,34].

In our cohort, in multivariate analysis neither reduced HRV alone nor depression provided significant prognostic information, which was exclusively linked to the reduced HRV comorbid with MDD. In ischemic heart disease, only few studies including small numbers of participants evaluated the prognostic value of HRV/MDD [35,36]. They concluded that reduced HRV represents “an independent mediator of the increased mortality observed in depressed patients” [36]. In our cohort, particularly moderate/severe symptoms of MDD were associated with prognosis. In similarity to our findings prior studies evaluating depressive patients without underlying cardiac diseases and with HF unrelated to ACHD reported a coherence between outcome and MDD severity [5,37].

MDD is linked to several comorbidities potentially associated with adverse cardiovascular outcome. This includes the dysregulation of the hypothalamus–pituitary–adrenal axis [38], ectopic fat compartments [39] and life style changes (particularly low physical activity, increased BMI and increased smoking behaviour) [40,41]. In particular, persisting MDD decreases adherence, contributing to an increased mortality in patients with acute coronary syndrome, thereby pointing to the importance of identifying MDD in the context of cardiovascular disorders [42]. In ACHD, non-adherence to scheduled outpatient-clinic appointments is associated with poorer survival, which may be partially associated with undiagnosed MDD [43].

HRV is considered to be a biomarker of the autonomic system. Our results point to an important heart and brain interaction in ACHD. Considering the observed impact of HRV/MDD on outcome, further studies are needed to evaluate the relation between HRV/MDD and long-time sequela of the underlying heart disease itself. This might help to understand the interaction between HRV and depression in ACHD.

In addition, we could demonstrate that prognostically relevant HRV information can be derived easily in an outpatient clinic during one visit by performing short-term analysis of HRV. Self-rating scales might be helpful to determine the diagnosis of MDD. Our study group evaluated the total Hospitality-Anxiety-Depression-Scale, the subscale of depression and the Beck-Depression-Inventory-II. These scales are very suitable to detect MDD with moderate/severe symptoms [30]. Future risk assessment of HRV added by the information of self-rating scales might provide an additional helpful tool for risk stratification in ACHD.

## 5. Conclusions

The data of the present study support the assumption that autonomic nervous system dysfunction measured as altered HRV is one of the pathways linking MDD and adverse outcomes. In particular, MDD with moderate to severe symptoms and advanced heart disease are related to these findings. Though the exact mechanism of the underlying pathophysiology still needs to be determined, screening for MDD and HRV measurement appear to be additional valuable tools for risk assessment in ACHD.

## Figures and Tables

**Figure 1 jcm-10-01554-f001:**
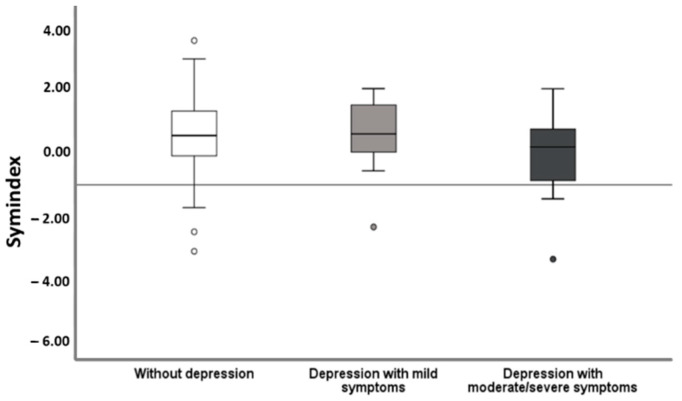
Heart rate variability in relation to severity of depressive symptoms. Patients with moderate/severe MDD presented with significantly lower HRV measured as the Symindex compared to those lacking depression (*p* = 0.008) and those with mildly symptomatic MDD (*p* < 0.049).

**Figure 2 jcm-10-01554-f002:**
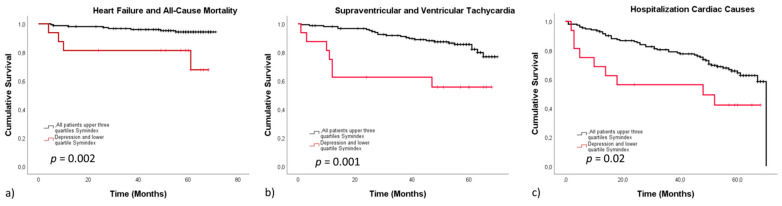
Survival stratified according to the lower quartile of the Symindex comorbid with MDD versus all other patients. The Kaplan–Meier curves display event free survival stratified according to the lower quartile of the Symindex comorbid with depression versus all other patients. Reduced heart rate variability associated with depression presented with a significantly lower event-free survival in all end-points, including heart failure and all-cause mortality (**a**), arrhythmias (**b**) and hospitalization due to cardiac causes (**c**).

**Table 1 jcm-10-01554-t001:** Underlying congenital heart disease.

Underlying Cardiovascular Disease	*n*
Tetralogy of Fallot	25
Atrioventricular septal defect	7
Simple shunts	9
Mild valve disease	7
Subaortic stenosis	4
Aortic valve disease (valve replacement, conduit, more than mild valve lesions)	18
Fontan circulation	12
D-Transposition of the great arteries: Atrial redirection surgery	17
Congenital pulmonary stenosis	7
Marfan	15
Coarctation	20
Eisenmenger syndrome	4
Congenitally corrected transposition	3
Ebstein anomaly	4
Miscellaneous	19
Total cohort	171

**Table 2 jcm-10-01554-t002:** Patients’ characteristics. Baseline characteristics of the total cohort and patients stratified according depressive state. Mild depression did not exhibit significantly reduced heart rate variability.

Parameter	Total Cohort (*n* = 171)	Without ^13^ MDD (Group a: *n* = 123)	Mild ^13^ MDD (Group b: *n* = 20)	Moderate/Severe ^13^ MDD (Group c: *n* = 28)	*p*-Value
**Age (years)**	35.6 ± 11.4	35.2 ± 11.7	36.1 ± 2.3	37.0 ± 11.0	^14^ n.s.
**Body Mass Index (kg/m^2^)**	25.3 ± 6.0	24.9 ± 6.4	25.9 ± 4.825.3	26.6 ± 4.9	n.s.
**Disease Complexity: Great/moderate/low (%)**	53.2/28.1/18.7	53.6/27.6/18.8	35/35/30.7	64.3/25/10.7	b/c < 0.001
**^1^ NYHA-Class I/II/(II/IV) (%)**	76.0/18.7/5.3	83.7/11.4/4.9	70/25/5	46.4/46.4/7.2	a/c < 0.001
**Male sex (n/%)**	73/42.7	47/61.3	11/55	15/53.6	n.s.
**Hypertension (%)**	29	20.2	36.8	14	n.s.
**Epicardial fat (cm)**	0.43 ± 0.18	0.39 ± 0.17	0.52 ± 0.18	0.55 ± 0.20	a/b = 0.005 a/c < 0.001
**^2^ HADS-D**	3.6 ± 3.8	2.1 ± 2.5	5.5 ± 2.2	9.1 ± 4.3	a/b < 0.001 a/c < 0.001 b/c < 0.001
**^3^ BDI-II**	7.6 ± 8.53	4.0 ± 4.8	11.7 ± 6.6	20.9 ± 8.0	a/b < 0.001 a/c < 0.001 b/c < 0.001
**Oxygen-saturation (%)**	98.3 ± 2.5	98.5 ± 2.1	97.4 ± 3.9	98.3 ± 2.5	n.s.
**^4^ HsCRP (mg/L)**	3.3 ± 6.1	3.2 ± 5.9	3.3 ± 6.3	3.8 ± 7.1	n.s.
**Creatinine (µmol/L)**	77.5 ± 15.6	77.0 ± 15.0	71.9 ± 13.9	83.5 ± 17.4	n.s.
**^5^ GGT (U/L)**	39.8 ± 60.3	39.5 ± 64.0	30.8 ± 21.9	47.6 ± 62.8	n.s.
**^6^ NT-proBNP (ng/L)**	182 ± 253	169 ± 208	99 ± 96	302 ± 428	c/a = 0.034 c/b = 0.018
**^7^ HBA1c (%)**	5.34 ± 0.34	5.31 ± 0.33	5.34 ± 0.42	5.45 ± 0.28	n.s.
**^8^ LDL(mg/dL)**	118.9 ± 31.2	116.4 ± 32.1	128.6 ± 23.2	122.9 ± 31.0	n.s.
**QRS (ms)**	120 ± 28	119 ± 28	109 ± 19	127 ± 33	n.s.
**^9^ QTc (ms)**	451 ± 31	451 ± 37	423 ± 21	436 ± 40	n.s.
**^10^ SDNN**	97.9 ± 47.4	101.0 ± 48.8	83.3 ± 34.7	94.1 ± 49.0	n.s.
**^11^ Triangular Index**	15.7 ± 8.6	16.3 ± 8.7	15.4 ± 7.3	13.5 ± 9.1	n.s.
**^12^ Symindex**	0.48 ± 1.12	0.59 ± 1.00	0.65 ± 1.06	−0.11 ± 1.45	c/a < 0.001 c/b = 0.057

^1^ New York Heart Association, ^2^ Hospital Anxiety and Depression Subscale, ^3^ Beck Depression Inventory, ^4^ high-sensitivity C-reactive protein, ^5^ gamma-glutamyl transferase, ^6^ N-terminal pro-B-type natriuretic peptide, ^7^ glycolized haemoglobin levels, ^8^ low density lipoproteins, ^9^ heart rate corrected QT-duration calculated with the Bazett formula, ^10^ standard deviation of the interbeat intervals, ^11^ geometric measure of the integral of the density of the interbeat interval histogram divided by its height, ^12^ logarithmically transformed heart rate variability (HRV) ratio derived from the normalized power spectral density of the low frequency band and the normalized power spectral density of the high frequency band, ^13^ Major depressive disorder, ^14^ not significantly different.

**Table 3 jcm-10-01554-t003:** Predictors of moderate/severe depression. Univariate binary regression analysis of baseline clinical characteristics identified single predictors of depression. Multivariate analysis included all parameters with a significance level <0.08. In multivariate analysis, Symindex and epicardial fat remained as the only independent predictors of moderate/severe depression. Mild depression was not associated with reduced heart rate variability.

Binary Logistic Regression Analysis	Without Depression Versus all Patients with Depression	*p*-Value	Without Depression Versus Mild Depressive Symptoms	*p*-Value	Without Depression/ Mild Depressive Symtoms Versus Moderate/ Severe Symptoms	*p*-Value	Without Depression Versus Moderate/ Severe Depressive Symptoms	*p*-Value
**Univariate analysis**	^9^ HR (95%^10^ CI)		^9^ HR (95%^10^ CI)		^9^ HR (95%^10^ CI)		^9^ HR (95%^10^ CI)	
**^1^ Symindex**	0.74 (0.55–0.99)	0.048	1.06 (0.66–1.70)	0.813	0.59 (0.41–0.85)	0.004	0.59 (0.40–0.86)	0.006
**Age**	1.01 (0.98–1.04	0.461	1.01 (0.97–1.05)	0.743	1.02 (0.98–1.05)	0.476	1.01 (0.98–1.05)	0.456
**Sex**	1.91 (0.97–3.75)	0.060	1.98 (0.76–51.3)	0.161	0.21 (0.75–3.82)	0.206	1.97 (0.82–4.27)	0.193
**^2^ NYHA-Class**	2.32 (1.31–4.10)	0.004	1.53 (0.71–3.34)	0.280	2.64 (1.42–4.90)	0.002	2.83 (1.49–5.41)	0.002
**Disease Complexity**	1.03 (0.67–1.58)	0.902	1.58 (0.89–2.83)	0.119	0.66 (0.37 1–18)	0.159	0.71 (0.39–1.27)	0.247
**Number of cardiac operations**	0.99 (0.73–1.35)	0.958	0.76 (0.48–1.22)	0.256	1.22 (0.85–1.75)	0.291	1.17 (0.81–1.67)	0.393
**Hypertension**	1.31 (0.59–2.87)	0.505	2.61 (0.97–7.08)	0.059	0.56 (0.18–1.72)	0.307	0.65 (0.21–2.06)	0.467
**Body Mass Index**	1.04 (0.98–1.10)	0.175	1.02 (0.96–1.09)	0.473	1.04 (0.98–1.10	0.226	1.04 (0.98–1.10)	0.206
**Epicardial fat**	123 (14.3–1061)	<0.0001	41.4 (3.19–537)	0.004	64.5 (6.23–666)	<0.0001	149 (11.7–1917)	<0.0001
**Oxygen-saturation (%)**	0.91 (0.80–1.03)	0.132	0.99 (0.80–1.25)	0.992	0.87 (0.76–0.99)	0.041	0.87 (0.76–0.99)	0.047
**^3^ HsCRP (mg/L)**	1.01 (0.96–1.06)	0.727	1.00 (0.93–1.08)	0.964	1.01 (0.96–1.08)	0.655	1.01 (0.96–1.08)	0.654
**Creatinine (µmol/L)**	1.01 (0.97–1.03)	0.531	0.97 (0.94–1.01)	0.156	1.03 (1.03–1.05)	0.029	1.03 (1.00–1.05)	0.053
**^4^ GGT (U/L)**	1.00 (0.99–1.01)	0.981	0.99 (0.98–1.01)	0.552	1.00 (0.99–1.01)	0.647	1.00 (0.97–1.01)	0.551
**^5^ NT-proBNP (ng/L** **)**	1.00 (0.99–1.00)	0.273	0.99 (0.99–1.00)	0.128	1.00 (1.00–1.01)	0.020	1.00 (1.00–1.01)	0.035
**^6^ HBA1c (%)**	2.24 (0.83–6.07)	0.133	1.31 (0.38–4.56)	0.672	2.67 (0.90–7.89)	0.076	3.17 (0.90–11.2)	0.072
**^7^ LDL(mg/dL)**	1.01 (0.99–1.02)	0.096	1.01 (0.99–1.03)	0.108	1.01 (0.96–1.02)	0.451	1.00 (0.99–1.02)	0.326
**QRS (ms)**	1.00 (0.99–1.01)	0.945	0.98 (0.96–1.01)	0.119	1.01 (0.99–1.02)	0.141	1.01 (0.99–1.02)	0.237
**^8^ QTC (ms)**	1.00 (0.99–1.01)	0.452	0.99 (0.98–1.11)	0.793	1.01 (0.99–1.02)	0.179	1.01 (0.99–1.02)	0.212
**Multivariate analysis**								
**^1^ Symindex**					0.63 (0.44–0.91)	0.014	0.62 (0.43–0.90)	0.013
**Epicardial fat**					48.8 (4.30–558)	0.004	131 (9.60–1798)	<0.0001

^1^ Logarithmically transformed HRV ratio derived from the normalized power spectral density of the low frequency band and the normalized power spectral density of the high frequency band, ^2^ New York Heart Association, ^3^ high-sensitivity C-reactive protein, ^4^ gamma-glutamyl transferase, ^5^ N-terminal pro-B-type natriuretic peptide, ^6^ glycolized haemoglobin levels, ^7^ low density lipoproteins, ^8^ corrected QT-duration calculated with the Bazett formula, ^9^ hazard ratio with 95% ^10^ Confidence Interval.

**Table 4 jcm-10-01554-t004:** Cox proportional survival analysis stratified according to combined parameter reduced heart rate variability (lower quartile of the Symindex) and depression. The data of univariate and multivariate Cox proportional Regression Analysis are presented separately for all three endpoints—heart failure and death, hospitalization and arrhythmias. The parameter Symindex comorbid with depression included depressive patients within the lower quartile of the Symindex. Even after adjustment for age, sex, and β-Blocker therapy, reduced heart rate variability remained as an independent predictor. In multivariate proportional survival analysis, the combined parameter of the lower quartile of the Symindex comorbid with depression remained as an independent predictor, whereas neither the lower quartile of the Symindex alone nor depression yielded significant differences.

Cox Proportional Regression Analysis	Heart Failure and All-Cause Mortality	*p*-Value	Hospitalization	*p*-Value	Supraventricular/Ventricular Tachycardia	*p*-Value
**Univariate analysis**	^9^ HR (95%CI)		^9^ HR (95%CI)		^9^ HR (95%CI)	
**^1^ Symindex (lower quartile)**	3.95 (1.25–12.5)	0.019	1.88 (1.11–3.17)	0.029	2.90 (1.43–5.87)	0.003
**^1^ Symindex (lower quartile) comorbid with depression**	5.30 (1.59–17.6)	0.006	2.27 (1.13–4.61)	0.024	3.80 (1.63–8.80)	0.002
**^2^ Major depressive disorder**	3.94 (1.25–12.4)	0.019	1.26 (0.66–2.43)	0.487	2.14 (0.96–4.80)	0.064
**Age**	1.00 (0.96–1.059	0.902	1.01 (0.99–1.04)	0.264	0.98 (0.99–1.04)	0.364
**Sex**	0.66 (0.20–2.20)	0.504	0.83 (0.49–1.40)	0.485	0.44 (0.20–0.99)	0.048
**^3^ NYHA-Class**	2.71 (1.34–4.55)	0.006	2.30 (1.59–3.32)	<0.0001	1.87 (1.12–3.12)	0.016
**Disease Complexity**	0.14 (0.02–0.96)	0.045	0.33 (0.20–0.59)	<0.0001	0.25 (0.11–0.58)	0.001
**Number of cardiac operations**	0.64 (0.26–1.59)	0.340	1.51 (1.17–1.95)	0.001	1.65 (1.20–2.27)	0.002
**Hypertension**	1.19 (0.32–4.39)	0.796	0.91 (0.48–1.72)	0.779	0.85 (0.35–2.08)	0.725
**Body Mass Index**	1.04 (0.98–1.10)	0.246	1.01 (0.98–1.05)	0.532	1.02 (0.98–1.07)	0.290
**Epicardial fat**	18.50 (1.24–276)	0.034	7.93 (2.03–31.0)	0.003	26.8(4.04–178)	0.001
**Oxygen-saturation (%)**	0.93 (0.78–1.10)	0.418	0.88 (0.81–0.95)	0.002	0.92 (0.82–1.03)	0.145
**^4^ HsCRP (mg/L)**	1.01 (0.96–1.09)	0.607	0.99 (0.96–1.03)	0.524	0.96 (0.88–1.05)	0.420
**Creatinine (µmol/L)**	1.03 (0.99–1.06)	0.059	1.00 (0.99–1.02)	0.847	1.03 (1.01–1.05)	0.006
**^5^ GGT (U/L)**	1.00 (0.99–1.01)	0.212	1.00 (1.00–1.01)	0.017	1.00 (0.98–1.01)	0.605
**^6^ NT-proBNP (ng/L** **)**	1.00 (1.00–1.00)	<0.0001	1.00 (1.00–1.00)	<0.0001	1.00 (1.00–1.01)	<0.0001
**^7^ HBA1c (%)**	1.82 (0.62–5.38)	0.280	1.38 (0.78–2.45)	0.265	1.97 (1.05–3.71)	0.035
**^8^ LDL(mg/dL)**	0.99 (0.98–1.02)	0.770	0.99 (0.99–1.01)	0.513	1.00 (0.99–1.01)	0.842
**QRS**	1.01 (0.99–1.03)	0.163	1.01 (1.01–1.02)	0.019	1.01 (0.97–1.02)	0.186
**QTc**	1.01 (0.96–1.03)	0.166	1.01 (1.00–1.02)	0.021	0.99 (0.98–1.01)	0.615
**Multivariate analysis**						
**^1^ Symindex (Lower Quartile) comorbid with depression**	7.04 (1.87–26.5)	0.004	3.8 (1.36–10.6)	0.011	4.90 (1.74–9.25)	0.003
**^6^ NT-proBNP**	1.00 (1.00–1.00)	<0.0001	1.00 (1.00–1.03)	0.010	1.00 (1.00–1.00)	<0.0001
**Disease complexity**			0.39 (0.22–0.66)	0.001	0.39(0.16–0.92)	0.032

^1^ Logarithmically transformed HRV ratio derived from the normalized power spectral density of the low frequency band and the normalized power spectral density of the high frequency band. ^2^ Moderate/severe versus MDD with mild depressive symptoms and lacking depression, ^3^ New York Heart Association, ^4^ high-sensitivity C-reactive protein, ^5^ gamma-glutamyl transferase, ^6^ N-terminal pro-B-type natriuretic peptide, ^7^ glycolized haemoglobin levels, ^8^ low density lipoproteins, ^9^ hazard ratio with 95% Confidence Interval.

## Data Availability

Data are not available.

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
