# Peer review of "Depression Associated with Reduced Heart Rate Variability Predicts Outcome in Adult Congenital Heart Disease"

_jcm, 2021, doi:10.3390/jcm10081554_

Round 1

Reviewer 1 Report

In this analysis, the authors report the frequency of mental depression (diagnosed by standardized interview) among 171 adults with congenital heart disease. They found a week association with heart rate variability. During a mean follow-up of 4 ½ years they explored the association of mental depression, heart rate variability and its combination on a broad variety of outcomes.

Although the study concerns a topic of interest, there may be some concerns about the presentation of data and the analysis performed.

Major comments

Baseline characteristics should be reported in more detail:

  • Type of heart defects (at least important categories, such as univentricular physiology, cyanotic heart disease, Eisenmenger syndrome, systemic right ventricles, etc.).
  • Systemic ventricular ejection fraction
  • Pulmonary hypertension
  • Exercise capacity
  • Anemia
  • etc.

Data analysis:

  • Without incorporating important variables with well established prognostic impact (ventricular function, exercise capacity, previous arrhythmia episodes, previous heart failure admissions, etc.) the analysis of outcomes is meaningless as it allows no conclusion about its importance as an independent predictor.
  • Given the presented data it is rather unlikely that mental depression and/or heart rate variability are independent predictors of outcome. Given its association with defect complexity it seems that these factors are an epiphenomenon of cardiac disease. This should be explored in much more detail. Other factors with impact of mental well-being, such as circumstances of living (living with partner, unemployment, major life events, etc.) should be reported.
  • I am convinced that you would create the very similar or even more impressive KM-curves when combining mental depression score with ejection fraction or with NT-proBNP or with previous heart failure admissions, etc.

Suggestion:

  • Rather than exploring the prognostic capability of mental depression, it may be more informative of exploring factors that are related to mental depression
  • What was your treatment algorithm for patients diagnosed with mental depression?
  • If heart rate variability is an important risk marker – then it may be very interesting to explore, whether treatment of mental depression by means of medication or other treatment were associated with changes of heart rate variability – was this explored in this study?

Minor comments:

  • Figure 1, Panel B: This is a very strange comparison – to compare to lowest quartile with the rest of the cohort (even when adding a second variable) will always generate a significant difference.
  • Table 2: Not all figures are legible
  • Epicardial fat is a very unusual risk marker in a cohort of young ACHD-patients
  • Unexpected hospital admission is a very broad endpoint

Author Response

Response to reviewer I

 Thank you very much for your detailed review.

Major comments

Baseline characteristics should be reported in more detail:

  • Type of heart defects (at least important categories, such as univentricular physiology, cyanotic heart disease, Eisenmenger syndrome, systemic right ventricles, etc.).

A table presenting the underlying congenital defects is provided.

  • Systemic ventricular ejection fraction

This is an interesting parameter, however our analysis included NT-proBNP, which at least is an excellent prognostic parameter in ACHD. Ventricular function and NT-proBNP levels show a close correlation. Therefore it does not make sense to include both parameters in one regression analysis. We decided to use NT-proBNP because NT-proBNP is the best biomarker reflecting prognosis in ACHD.

In our cohort  only 4 patients presented with a history of decompensation  Therefore we did not include this parameter in  our analysis.

  • Pulmonary hypertension

Pulmonary hypertension represents a frequent long-term complication in ACHD. We cannot provide hemodynamic data. The group of Eisenmenger patients only included 4 patients, thus this group was too small for or a separate analysis.

  • Exercise capacity

Our analysis did not include exercise capacity, because 15% of patients did not perform cardiorespiratory function test. These patients were characterized by higher NYHA-class and a higher prevalence of MDD. Exclusion of these patients refers to a potential selection bias towards participants with preserved functional capacity and exclusion of participants with MDD.

  • Anemia

Anemia was not associated with MDD or outcome parameters.

Data analysis:

  • Without incorporating important variables with well-established prognostic impact (ventricular function, exercise capacity, previous arrhythmia episodes, previous heart failure admissions, etc.) the analysis of outcomes is meaningless as it allows no conclusion about its importance as an independent predictor.

We disagree that we did not include prognostic relevant parameters reflecting adverse outcome of the underlying heart defect. More details are provided in the Method section (Statistics).

As outlined above the incorporated variable NT-proBNB represents the best prognostic parameters regarding cardiovascular outcome in ACHD. Moreover NT-proBNP and ventricular function show a close correlation.

  • Given the presented data it is rather unlikely that mental depression and/or heart rate variability are independent predictors of outcome. Given its association with defect complexity it seems that these factors are an epiphenomenon of cardiac disease. This should be explored in much more detail. Other factors with impact of mental well-being, such as circumstances of living (living with partner, unemployment, major life events, etc.) should be reported.

We disagree.  In univariate analysis MDD and HRV were single preditcors. We reported, that in multivariate analysis only the combination of MDD and HRV remained significant predictors. As discussed similar observations have been made in non-congenital heart disease. Surprisingly we found similarities despite the differences in underlying cardiac disease,

It is a good idea to incorporate the impact of life events on HRV. Unfortunately we cannot provide data.

You think that the association between outcome and MDD/HRV might be an epiphenomenon of the underlying cardiac situation. This may be totally true.  However, it is unknown. It still has to be discussed whether MDD is a causative factor of cardiac deterioration or the other way round. Independentlöy from its origin, the combined presence of MDD and reduced HRV ist associated wiith adverse outcome.    

  • I am convinced that you would create the very similar or even more impressive KM-curves when combining mental depression score with ejection fraction or with NT-proBNP or with previous heart failure admissions, etc.

We are sure, that survival curves might be impressive using NT-proBNP. However, this is already known. Our aim was to analyse the potential impact of depression on outcome. At present there are three available studies reporting a potential relationship between MDD and outcome in ACHD. There is only one prospective study including 60 patients.

As already mentioned, decompensation was very rare in our cohort. This information is provided in the manuscript.This analysis is not meaningful.

Suggestion:

  • Rather than exploring the prognostic capability of mental depression, it may be more informative of exploring factors that are related to mental depression

This is already presented in the manuscript in table 3 (initially table 2)

  • What was your treatment algorithm for patients diagnosed with mental depression?

Treatment of MDD did not follow a specific protocol. Our mental health professional (KGK) recommended treatment including medical therapy and/or psychotherapy dependent on clinical aspects and .  However, the answer regarding optimal therapy requires an adequately powered prospective study with careful assessment of MDD at given intervals. This question was not in the focus of the present study. Future studies also should include changes of heart rate variability assessment.

  • If heart rate variability is an important risk marker – then it may be very interesting to explore, whether treatment of mental depression by means of medication or other treatment were associated with changes of heart rate variability – was this explored in this study?

Unfortunately this was not evaluated. 

Minor comments:

  • Figure 1, Panel B: This is a very strange comparison – to compare to lowest quartile with the rest of the cohort (even when adding a second variable) will always generate a significant difference.

We decided to include this figure to demonstrate the differences of heart rate variability in the different groups. We think this visualisation is helpful to realize the dimensions of this previously not described parameter

  • Table 2: Not all figures are legible.

I am sorry. Originally as well as in the present version the tables can be seen in  detail.

  • Epicardial fat is a very unusual risk marker in a cohort of young ACHD-patients

In ACHD epicardial fat is increased in patients with MDD. This was published previously, which to our opinion justifies the inclusion in the analysis. The literature is provided.

  • Unexpected hospital admission is a very broad endpoint

We agree that unexpected hospital admission is a very broad endpoint. However the present literature includes this endpoint. Therefor we also analysed our data according this question.

Reviewer 2 Report

This is a prospective cohort study including 171 adult congenital heart disease (ACHD) patients who underwent who participated in the heart rate variability (HRV) measurements. This is a subgroup of a larger study that assessed mental illness in ACHD patients. Although, in multivariate analysis neither HRV or major depressive disorder (MDD) were predictive of poor outcomes, a combination of HRV with MDD predicted poor outcomes.

Overall, this is a nicely designed study and has a considerable number of patients considering prospective study design. I congratulate authors on their nice work. However, the manuscript should be made more concise and to the point focusing on the major theme of the study.

Major Comments:

  • Suggest rephrasing the conclusion in the abstract.
  • Suggest rephrasing the introduction and combining multiple small paragraphs into one theme instead of scattered ideas. Please provide more information on HRV as it is a newly described finding.
  • How was complexity of ACHD classified?
  • Can you provide details of the patients who had worse outcomes in terms of congenital heart defect, palliation or surgical status? Were there any single ventricle patients? Were there patients with multiple surgeries?
  • How many patients were on beta blocker therapy?
  • How many patients were undergoing the treatment with antidepressants or therapy?
  • Was the HRV measurements obtained near the psychological assessment? If the HRV was obtained farther from the psychiatric assessment, it may not accurately represent the MDD status or severity.
  • What was the psychiatric status of patients towards the study end-point or follow up? How long was the follow up? How many times was the HRV obtained?
  • Is there a validation cohort?
  • How was the epicardial fat calculated? There was quite a significant variability in the epicardial fat measurement, was median or mean values considered for analysis? Can authors comment on it in the discussion.
  • Page 6, line 172. Hospitalization is not-significant with a p value of 0.068 which is not near significance so recommend rephrasing the sentence.
  • Page 8, line 227. Recommend rephrasing near significant to NOT significance.

Minor Suggestions:

  • Introduction, sentence on page 1 - 37, 38 and 39 can be improved. Please provide more details on individual elements. E.g. Depression has been shown to predict worse long-term cardiovascular outcomes among patients with atrial fibrillation and heart failure.
  • Comma after in particular on page 1, line 42.
  • What does SDNN stands for? Please mention abbreviation in the table 1 footnotes.

Author Response

Response to reviewer II

Thank you very much for your detailed review.

Major Comments:

  • Suggest rephrasing the conclusion in the abstract.

The abstract including the conclusion has been rewritten

  • Suggest rephrasing the introduction and combining multiple small paragraphs into one theme instead of scattered ideas. Please provide more information on HRV as it is a newly described finding.

A paragraph describing heart rate variability has been included. We think the iintruduction is fine, we would like to keep it as it is.

  • How was complexity of ACHD classified?

Complexity of heart disease was assessed according the Warnes classification, the information as well as the literature is included

  • Can you provide details of the patients who had worse outcomes in terms of congenital heart defect, palliation or surgical status? Were there any single ventricle patients? Were there patients with multiple surgeries?

A table is included depicting the underlying congenital defects. We included the operations in binary regression and Cox proportional survival analysis.

  • How many patients were on beta blocker therapy?

The data are included in the manuscript

  • How many patients were undergoing the treatment with antidepressants or therapy?

The data are included in the manuscript

  • Was the HRV measurements obtained near the psychological assessment? If the HRV was obtained farther from the psychiatric assessment, it may not accurately represent the MDD status or severity. Detailed description is provided in the method section.
  • What was the psychiatric status of patients towards the study end-point or follow up? How long was the follow up? How many times was the HRV obtained?

Unfortunately we did not repeat psychological assessment during this follow-up. HRV measuremnts was only obtained at baseline. Mean follow-up was 54.7±14.9 months .

  • Is there a validation cohort?

Unfortunately not.

  • How was the epicardial fat calculated? There was quite a significant variability in the epicardial fat measurement, was median or mean values considered for analysis? Can authors comment on it in the discussion.pkay
  • Echocardiographic assessment of epicardial fat was derived from the parasternal long/short axis view at end diastole. Measurments of epicardial fat thickness was derived from the free right ventricular wall perpendicular to the aortic annulus.We used mean values. The selection of the variables is provided in the manuscript. We included parameters associated with depression as well as accepted characteristics of adverse outcome in adult congenital heart disease.Details are provided in “statistica analysis.
  • Page 6, line 172. Hospitalization is not-significant with a p value of 0.068 which is not near significance so recommend rephrasing the sentence.

This has been changed.

  • Page 8, line 227. Recommend rephrasing near significant to NOT significance.okay
  • This has been changed

Minor Suggestions:

  • Introduction, sentence on page 1 - 37, 38 and 39 can be improved. Please provide more details on individual elements. E.g. Depression has been shown to predict worse long-term cardiovascular outcomes among patients with atrial fibrillation and heart failure.

A general description of HRV is included in the Introduction section, which to our opinion is fine. We would like to keep iit as it is.

  • Comma after in particular on page 1, line 42.

This has been changed.

  • What does SDNN stands for? Please mention abbreviation in the table 1 footnotes.

The explanation is provided in the foot notes of the table.

Round 2

Reviewer 1 Report

The authors have tried to address the reviewer’s concerns and comments. In some points, there is disagreement between reviewers and authors, which is OK. Other points could not be resolved due to lack of data.

Although doubts remain about the general validity of the results of the paper, the work certainly stimulates hypotheses that may be worth further exploration in the future.

Specific comments:

Figure 1, Panel B: Again, this is a very strange comparison that doesn’t make sense in my opinion

Conclusions within the manuscript diverge from conclusions within the abstract. This should be revised (conclusions within the abstract are more appropriate).

Author Response

The present study included established parameter of risk assessment in ACHD. In so far, we are convinced that the present study gives evidence, that alteration of HRV representing the biomarker of the autonomous nervous system, HRV,  provides valuable information. 

Figure 1, Panel B: Again, this is a very strange comparison that doesn’t make sense in my opinion

Answer: We reduced it to the half

Conclusions within the manuscript diverge from conclusions within the abstract. This should be revised (conclusions within the abstract are more appropriate).

Answer: The conclusion has been rewritten.